# Learning from Shortcut: A Shortcut-guided Approach for Graph Rationalization

## Abstract

The remarkable success in graph neural networks (GNNs) promotes the Graph Rationalization methods that aim to provide explanations to support the prediction results by identifying a small subset of the original graph (i.e., *rationale*). Although existing methods have achieved promising results, recent studies have proved that these methods still suffer from exploiting shortcuts in the data to yield task results and compose rationales. Different from previous methods plagued by shortcuts, in this paper, we propose a Shortcut-guided Graph Rationalization (SGR) method, which identifies rationales by learning from shortcuts. Specifically, SGR consists of two training stages. In the first stage, we train a *shortcut guider* with an early stop strategy to obtain shortcut information. During the second stage, SGR separates the graph into the rationale and non-rationale subgraphs and lets them learn from the shortcut information generated by the frozen *shortcut guider* to identify which information belongs to shortcuts and which does not. Finally, we employ the non-rationale subgraphs as environments and identify the invariant rationales which filter out the shortcuts under environment shifts. Extensive experimental results on both synthetic and real-world datasets clearly validate the effectiveness of our proposed method. Code is released at `https://anonymous.4open.science/r/codes-of-SGR-1340`.

## 1 Introduction

Graph neural networks (GNNs) have become ubiquitous in various applications exhibiting high performance (Kipf & Welling, 2017; Xu et al., 2019). One of the main application categories is the graph classification task, such as molecular graph property prediction (Hu et al., 2020; Guo et al., 2021; Yehudai et al., 2021). Despite their success, GNNs on graph classification tasks still suffer from a lack of explainability and reliability in their prediction results, which has prompted many researchers (Ying et al., 2019; Luo et al., 2020) to investigate how to provide explanations for GNNs. Among them, graph rationalization methods (Lei et al., 2016; Wang et al., 2021) have achieved increasing attention. These methods aim to yield the task results while identifying a small subset of the original graph (i.e., the *rationale*), such as significant nodes or edges. In this way, the extracted rationale can serve as an explanation for the prediction results.

Despite the appeal of graph rationalization methods, recent studies (Chang et al., 2020; Wu et al., 2022) have indicated that these approaches are susceptible to exploiting shortcuts (aka, spurious correlations) in the data to yield task results and compose rationales. Such exploitation can result in invalid or erroneous conclusions, undermining the reliability of the model's outputs.

Considering Figure 1, we predict the motif type based on the graph that consists of motifs and bases subgraphs. In the training dataset, the *Cycle*-motifs are frequently co-occurring with the *Tree* bases and *House*-motifs are predominantly accompanied by the *Wheel* bases, which may mislead the GNNs over-reliance on these associations for achieving high accuracy, rather than discerning the true relationships between critical subgraphs (i.e., *rationales*) and the predicted labels. For example, GNNs may predict the motif type as *Cycle* when identifying the *Tree* bases or classify the motif type as *House* when recognizing the *Wheel* bases. However, this dependency on biases can result in inaccuracies when facing out-of-distribution (OOD) data (the test dataset in Figure 1), such as incorrectly predicting a *Cycle*-motif with *Wheel* bases as a *House* or misclassifying *House*-motifs with *Tree* bases as *Cycles*.

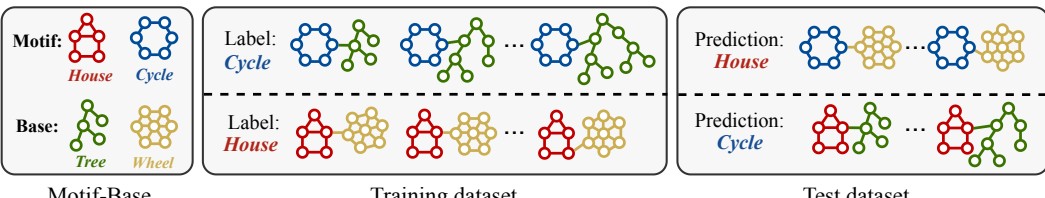

Figure 1: An example of the motif type prediction, where the *Cycle* and *House* are motif labels, and *Tree* and *Wheel* are bases that are irrelevant to the motif prediction. In the training dataset, the data distribution is *Cycle* with *Tree* and *House* with *Wheel*. When the model depends too much on this data distribution (i.e. shortcuts) for prediction, the model is likely to misclassify when facing the test dataset with a shift in the distribution.

To solve that, various methods (Fan et al., 2022; Sui et al., 2022; Li et al., 2022b) have been proposed recently to compose the real rationale by capturing the invariant relationship between rationales and their labels. These methods argue that the rationale behind the labels remains stable across different environments. Therefore, they employ environment inference methods to obtain various latent environments, and then identify the invariant rationales under environment shifts.

The methods mentioned above are all based on the assumption that shortcuts are unknown. However, a direct approach is to explicitly identify which nodes in the graph are shortcuts, enabling us to use these shortcut nodes to train a de-biased model. Unfortunately, annotating nodes for shortcuts on each graph can be a laborious task. Interestingly, although obtaining shortcut nodes is unavailable, we can get latent shortcut representations. Research (Clark et al., 2019; Nam et al., 2020; Li et al., 2021; Fan et al., 2022) has demonstrated that shortcut features are easier to learn than rationale features, indicating the features learned by the model in initial training stages are more inclined to shortcuts (Arpit et al., 2017). Therefore, we can obtain the shortcut representations with an early stop strategy.

Along this line, in this paper, we propose a **S**hortcut-guided **G**raph **R**ationalization (SGR) method, which identifies significant nodes as rationales by learning from shortcuts. Specifically, our method involves two stages. In the first stage, we train a *shortcut guider* which is designed to intentionally capture the shortcut in data with the early stop strategy. In the second stage, we first freeze the trained *shortcut guider* and adopt it to generate the shortcut representation. Then, we separate the original input graph into rationale and non-rationale subgraphs, which are respectively encoded into representations. Next, we employ the *shortcut guider* to eliminate the shortcut information from the rationale subgraphs by minimizing the Mutual Information (MI) (Poole et al., 2019; Cheng et al., 2020a; Yue et al., 2022) between the shortcut and rationale representation. Meanwhile, we also let the *shortcut guider* encourage the non-rationale subgraphs and shortcut representations to encode the same information by maximizing MI (Oord et al., 2018). Based on the MI estimation methods, rationale and non-rationale subgraph representations can fully learn which information belongs to shortcuts and which does not. Finally, to further identify the invariant rationales under environment shifts, we consider non-rationale representations which sufficiently capture the shortcut information as the environment. We then combine each rationale representation with various non-rationale representations, and encourage these combinations to maintain a stable prediction and yield rationales. Experiments over ten datasets, including various synthetic (Ying et al., 2019; Wu et al., 2022) and OGBG (Hu et al., 2020) benchmark datasets, validate the effectiveness of our proposed SGR.

## 2 SHORTCUT-GUIDED GRAPH RATIONALIZATION

### 2.1 PROBLEM DEFINITION

Considering graph classification tasks, given an input graph instance $g = (\mathcal{V}, \mathcal{E})$ with $N$ nodes and $Z$ edges and its graph-level ground truth $y$, where $(g, y) \in \mathcal{D}_G$, $\mathcal{D}_G$ is the dataset, $\mathcal{V}$ is the set of nodes, $\mathcal{E}$ is the set of edges and the matrix $A \in \{0,1\}^{|\mathcal{V}| \times |\mathcal{V}|}$, our goal is first to yield a rationale mask vector $\mathbf{M} \in \mathbb{R}^N$ that represents the probability of each node being selected as the rationale. Then, the of rationale subgraph is calculated as $\mathbf{h}_r = \text{READOUT}\left(\mathbf{M} \odot \text{GNN}_g(g)\right)$, where $\text{GNN}_g(\cdot)$ can be any GNN encoder (e.g., GIN (Xu et al., 2019)). Finally, the rationale representation $\mathbf{h}_r$ is employed to yield task results. Take the case in Figure 1 for example, our goal is to predict the motif type while identifying the *Cycle* or *House* structure as the rationale to support the prediction results.

## 2.2 ARCHITECTURE OF SGR

To explicitly utilize the shortcut information to compose unbiased rationales, we propose the SGR method consisting of two stages. In the first stage, we employ an early stopping strategy to obtain a *shortcut guider* that can fully learn the shortcut information. In the second stage, as shown in Figure 2, our method involves a *shortcut guider*, *selector*, and *predictor*. Initially, we freeze the *shortcut guider* and further obtain the shortcut representation. We then adopt the *selector* to separate the original graph into rationale and non-rationale representations. Next, we use

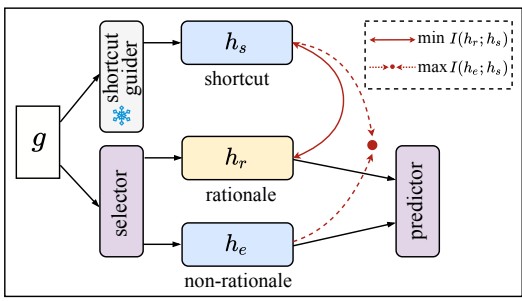

Figure 2: Architecture of SGR in the second stage.

the MI estimation method to transfer the generated shortcut information to both the rationale and non-rationale representation, ensuring these representations can learn from shortcuts. Finally, the *predictor* yields prediction results based on the above rationale and non-rationale representation cooperatively.

### 2.2.1 SHORTCUT GUIDER

Although it is difficult to identify which nodes are shortcuts, we assume that shortcut representations are available. Specifically, previous research (Li et al., 2021; Nam et al., 2020; Fan et al., 2022) suggests that shortcut features are easier to learn than rationale features, indicating that the features learned in the initial training stages are more inclined to shortcuts (Arpit et al., 2017). Therefore, in the first stage, we intentionally train the *shortcut guider* to capture the shortcut information with an early stop strategy. Initially, we train the *shortcut guider* on the dataset $\mathcal{D}_G$ to predict the graph label:

$$\mathbf{H}_s = \text{GNN}_s(g), \quad \mathbf{h}_s = \text{READOUT}\left(\mathbf{H}_s\right), \quad \hat{y}_s = \Phi_s\left(\mathbf{h}_s\right). \quad (1)$$

Among them, $\text{GNN}(\cdot)$ can be any GNN encoder such as GCN (Kipf & Welling, 2017). $\mathbf{H}_s \in \mathbb{R}^{N \times d}$ denotes the node representation, and $\mathbf{h}_s \in \mathbb{R}^d$ is the graph-level representation which is generated by a readout operator (employing mean pooling in this paper). $\Phi_s(\cdot)$ is a classifier which is applied to project $\mathbf{h}_s$ to the graph label. Then, the prediction loss can be formulated as:

$$\mathcal{L}_s = \mathbb{E}_{(g,y)\sim\mathcal{D}_G}\left[l(\hat{y}_s, y)\right], \quad (2)$$

where $l(\cdot)$ is the cross entropy loss. We then train the *shortcut guider* only for a few epochs (e.g., 3 epochs) to ensure the *shortcut guider* capture more shortcut information rather than rationale information. Finally, we freeze the parameters of the *shortcut guider* and apply this guider to the second stage.

### 2.2.2 SELECTOR

To separate the original input into rationale and non-rationale subgraphs, the *selector* first generates $\mathbf{M} \in \mathbb{R}^N$ that represents the probability of each node being selected as the rationale (Liu et al., 2022):

$$\mathbf{M} = \sigma(\Phi_m\left(\text{GNN}_m(g)\right)), \quad (3)$$

where $\Phi_m(\cdot)$ encodes each node into a value of selecting the node as the rationale, and $\sigma$ denotes the sigmoid function, indicating the probability of nodes being the rationale. Then, the *selector* employs another GNN encoder to obtain the node representation $\mathbf{H}_g = \text{GNN}_g(g)$. Next, the rationale node representation can be defined as $\mathbf{M} \odot \mathbf{H}_g$, while the non-rationale node representation is formulated as $(1 - \mathbf{M}) \odot \mathbf{H}_g$. Finally, the rationale subgraphs representation $\mathbf{h}_r$ and the non-rationale ones $\mathbf{h}_e$ can be obtained by a READOUT operation:

$$\mathbf{h}_r = \text{READOUT}\left(\mathbf{M} \odot \mathbf{H}_g\right), \quad \mathbf{h}_e = \text{READOUT}\left((1 - \mathbf{M}) \odot \mathbf{H}_g\right). \quad (4)$$

### 2.2.3 LEARNING FROM SHORTCUT BY MI ESTIMATION

To eliminate the shortcut information in the rationale and alleviate the problem of employing shortcuts in the data for prediction, we adopt the *shortcut guider* to reduce the mutual information between rationale subgraphs representations and shortcut representations. To achieve this, we first input the original graph $g$ into the *selector* to obtain the subgraphs representations $\mathbf{h}_r$ and $\mathbf{h}_e$, respectively,

as described in section 2.2.2. We then keep the *shortcut guider* frozen and employ it to generate shortcut representations $\mathbf{h}_s$. Next, we employ the MI minimization method to ensure that the shortcut information can be removed from the rationale (i.e., $\min I(\mathbf{h}_r; \mathbf{h}_s)$), where $I(;)$ denotes the MI, and MI is a measure of the mutual dependence between the two variables.

Meanwhile, we employ the MI maximization method to facilitate the matching of non-rationale representations with shortcut representations (i.e., $\max I(\mathbf{h}_e; \mathbf{h}_s)$), with the goal of enabling the full learning of shortcut information. Then, we consider the matched non-rationale representations as the environment and apply them to the *predictor*. Finally, the objective of learning from shortcut is:

$$\mathcal{L}_{shortcut} = I(\mathbf{h}_r; \mathbf{h}_s) - I(\mathbf{h}_e; \mathbf{h}_s). \tag{5}$$

In the implementation, we adopt CLUB_NCE (Yue et al., 2022) to achieve MI minimization, where CLUB_NCE is a variant of CLUB (Cheng et al., 2020a) that is designed to estimate the upper bound of MI. For MI maximization, we employ the InfoNCE (Oord et al., 2018) method. A detailed description of CLUB_NCE and InfoNCE can be found in Appendix A.

### 2.2.4 PREDICTOR

In the *predictor*, we first adopt the rationale representation to predict the graph label with the cross entropy loss $\mathcal{L}_r = \mathbb{E}_{(g,y) \sim \mathcal{D}_G} [l(\hat{y}_r, y)]$, where $\hat{y}_r = \Phi_p(\mathbf{h}_r)$ and $\Phi_p(\cdot)$ is a shared classifier. Afterward, to obtain the invariant rationales under environment shifts, we consider the non-rationale representations as environments and combine each rationale representation with various environments (Liu et al., 2022; Fan et al., 2022). Specifically, we transfer a batch of sample pairs $\left\{ \left( g^i, y^i \right) \right\}_{i=1}^{K}$ to their representations $\left\{ \left( \mathbf{h}_r^i, \mathbf{h}_e^i, y^i \right) \right\}_{i=1}^{K}$. After that, since the non-rationale (i.e., environment) does not affect the task prediction, we combine each rationale representation $\mathbf{h}_r^i$ with all non-rationale representation $\mathbf{h}_e^j$ ($\mathbf{h}_e^j \neq \mathbf{h}_e^i$) in the same batch to achieve environment shifts:

$$\mathbf{h}^{i,j} = \mathbf{h}_r^i + \mathbf{h}_e^j. \tag{6}$$

Meanwhile, the corresponding labels are unchanged since the rationale information in the synthetic data is unchanged. Next, we feed the combined graph representations to the shared classifier $\Phi_p(\cdot)$ to yield the task results, and the loss is calculated by the cross entropy function:

$$\hat{y}^{i,j} = \Phi_p(\mathbf{h}^{i,j}), \quad \mathcal{L}_e = \mathbb{E}_i \left[ \mathbb{E}_j \left[ l(\hat{y}^{i,j}, y) \right] \right]. \tag{7}$$

Finally, to make the predictions stable across different environments and mitigate the instability of the prediction results between the augmented data and the original data due to environmental changes, we first measure differences between $\hat{y}_r^i$ and $\hat{y}^{i,j}$ (i.e., $\mathcal{D}_f(\hat{y}_r^i; \hat{y}^{i,j})$), where $\mathcal{D}_f(\cdot)$ can be any distance function, such as squared euclidean distance). Then, to align the predicted distributions across environments with those predicted using rationale representations, we minimize the mean and variance of the differences:

$$\mathcal{L}_{diff} = \mathbb{E}_i \left[ \mathbb{E}_j \left[ \mathcal{D}_f(\hat{y}_r^i; \hat{y}^{i,j}) \right] + \text{Var}_j \left[ \mathcal{D}_f(\hat{y}_r^i; \hat{y}^{i,j}) \right] \right]. \tag{8}$$

### 2.3 TRAINING AND INFERENCE

During training, to encourage the model to control the expected size of rationale subgraphs, following Liu et al. (2022), we add a sparsity constraint on the probability $\mathbf{M}$ of being selected as rationale:

$$\mathcal{L}_{sp} = \left| \frac{1}{N} \sum_{i=1}^{N} M_i - \alpha \right|, \tag{9}$$

where $\alpha \in [0, 1]$ is the predefined sparsity level. Finally, the objective of SGR in the second stage is:

$$\mathcal{L}_{sgr} = \mathcal{L}_r + \mathcal{L}_e + \lambda_{diff}\mathcal{L}_{diff} + \lambda_{shortcut}\mathcal{L}_{shortcut} + \lambda_{sp}\mathcal{L}_{sp}, \tag{10}$$

where $\lambda_{diff}$, $\lambda_{shortcut}$ and $\lambda_{sp}$ are hyperparameters. At inference time, only $\mathbf{h}_r$ is employed to yield the task results.

## 3 EXPERIMENTS

In this section, to verify the reasonableness and effectiveness and of SGR, we first conduct experiments to validate that shortcut information is captured during the early stage of model training. Then, we compare SGR with several baseline methods on both synthetic and real-world datasets. Finally, we present visualizations showing the rationale subgraphs identified by SGR, which serves to provide further insight into the model's decision-making process.

### 3.1 DATASETS

Here, we make experiments on four synthetic datasets and six real-world benchmark datasets to evaluate the performance of our proposed approach for graph rationalization. Details of dataset statistics are summarized in Appendix B.2.

- **Spurious-Motif** (Ying et al., 2019; Wu et al., 2022) is a synthetic dataset for predicting the motif category of each graph. Each graph consists of two subgraphs, the motif subgraph (*Cycle*, *House*, *Crane* denoted by $R = 0, 1, 2$, respectively) and the base one (*Tree*, *Ladder*, *Wheel* denoted by $E = 0, 1, 2$ respectively). Among them, the motif subgraph is regarded as the ground-truth explanation (i.e. rationale) for the graph label, which suggests the graph label is solely determined by the motif subgraph. The base subgraph can be considered as the non-rationale (or environment). To verify the effectiveness of SGR, we manually generate several datasets containing shortcuts.

  Specifically, we construct the training dataset by sampling each motif uniformly, while controlling the distribution of the base through $P(E) = b \times \mathbb{I}(E = R) + \frac{1-b}{2} \times \mathbb{I}(E \neq R)$, where the degree of spurious correlation is controlled by $b$. In this paper, we set $b = \{0.5, 0.7, 0.9\}$. Besides, to verify the shortcut whether will be captured in the initial training stages, we first create a balance dataset (i.e., $b = \frac{1}{3}$, where each motif contains 1,000 training instances, for a total of 3,000 instances.). Then, based on this balance dataset, we intentionally conduct additional 1,000 instances that are all *Cycle* motifs with *Tree* bases, achieving the spurious correlations in *Cycle-Tree*. In the test dataset, we match the motif and base randomly ($b = \frac{1}{3}$) to construct an unbiased test dataset.

- **Graph-SST2** (Socher et al., 2013; Yuan et al., 2022) is a text sentiment analysis dataset, where each text instance in SST2 is converted to a graph. Following (Wu et al., 2022; Fan et al., 2022), to create distribution shifts, we divide the graphs into different sets according to their average node degrees, where the node degrees in the training set are higher than degrees in the test set.

- **Open Graph Benchmark (OGBG)** (Hu et al., 2020) is a benchmark dataset for machine learning on graphs, where we consider five OGBG-Mol datasets which are all employed for molecular property prediction (i.e. MolHIV, MolToxCast, MolBACE, MolBBBP and MolSIDER). We split datasets by default, where each split contains a set of scaffolds different to each other.

### 3.2 COMPARISON METHODS

First, we compare SGR with classical GNNs methods GCN (Kipf & Welling, 2017) and GIN (Xu et al., 2019). We then compare several competitive baselines specifically designed for explainable GNNs:

- **DIR** (Wu et al., 2022) discovers invariant rationales by separating the graph as the rationale subgraphs and the non-rationale ones. Different from SGR, DIR explicitly creates multiple environments by employing the non-rationale subgraphs.

- **DisC** (Fan et al., 2022), **GREA** (Liu et al., 2022) and **CAL** (Sui et al., 2022) all compose rationales by taking non-rationale subgraphs representations as environments. Differently, DisC selects edges as rationales, GREA identifies nodes as rationales, and CAL considers both edges and nodes.

- **GSAT** (Miao et al., 2022) learns stochasticity-reduced attention to select rationales based on the information bottleneck principle (Tishby et al., 2000; Alemi et al., 2017).

- **DARE** (Yue et al., 2022) proposes a disentanglement-augmented method to extract rationales. Meanwhile, it introduces CLUB_NCE to improve MI minimization. Although DARE is designed for explaining natural language understanding tasks. It can naturally be applied to explain GNNs.

Besides, in our experiments, we implement all of these explainable baselines with GCN (Kipf & Welling, 2017) and GIN (Xu et al., 2019) as the graph encoder, respectively.

Figure 3: (a) Accuracy of unbiased/biased test sets with different biases. (b) Curve of training losses on balance/bias examples of *Cycle-Tree*. (c) Training accuracy on balance/bias examples of *Cycle-Tree*. (d) Performance of SGR with different shortcut guiders that are trained with the early stop strategy.

## 3.3 EXPERIMENTAL SETUP

**Metrics.** In this paper, following the metric setting of Wu et al. (2022); Fan et al. (2022), we employ ACC to evaluate the task prediction performance for Spurious-Motif and Graph-SST2, ROC-AUC for OGBG. Besides, since the Spurious-Motif dataset contains ground-truth rationales, we adopt the Precision@5 metric to evaluate the difference between predicted rationales and real rationales.

**Optimization and Hyperparameters.** In all the experiments, we set $\lambda_{diff}$, $\lambda_{shortcut}$ and $\lambda_{sp}$ as 0.1, 0.01 and 1.0, respectively. The hidden dimensionality $d$ is set as 32 for Spurious-Motif, 64 for Graph-SST2, and 128 for OGBG. The learning rate of the Adam optimizer (Kingma & Ba, 2014) is initialized as 1e-2 for both Spurious-Motif and Graph-SST2, and 1e-3 for OGBG. We set the predefined sparsity $\alpha$ as 0.1 for MolHIV, 0.5 for MolSIDER, MolToxCast and MolBBBP, and 0.4 for other datasets. The early stop epoch is 2 or 3 for Spurious-Motif and 3 or 4 for both Graph-SST2 and OGBG. We employ the squared euclidean distance as $\mathcal{D}_f(\cdot)$. All methods are trained with five different random seeds on a single A100 GPU, and we report the test performance (with the mean results and standard deviations) of the epoch that achieves the best validation prediction performance.

## 3.4 EXPERIMENTAL RESULTS

**Do GNNs learn shortcuts during the initial training?** Since we fail to obtain which nodes are shortcuts explicitly, we assume shortcut features are easier to learn than the rationale ones, and we employ an early stopping strategy to get the shortcut information. In this section, we conduct serval experiments to validate our assumption. First, Arpit et al. (2017); Nam et al. (2020) have empirically proved that "*If the malignant bias is easier to learn than the real relationship between the input and label, the neural network tends to memorize it first.*".

As a type of neural networks, we argue that GNNs also obey this conclusion. Therefore, in line with this conclusion, we propose that by demonstrating shortcuts are malignant biases and shortcuts is easier to learn than real relationship between the input and label, we can infer these shortcuts are more likely to be captured during the initial training. To this end, we conduct experiments on Spurious-Motif. In addition to the unbiased test set, we construct biased test sets with degrees of bias $b$ matching those of the corresponding training sets. Then, we perform GCN on these datasets. As shown in Figure 3(a), we find GCN achieves promising results (Accuracy almost to 100%) on biased test set. However, when evaluating on the unbiased test set, the performance of GCN exhibits a significant degradation. These observations imply that the shortcuts in Spurious-Motif is malignant. Besides, as the degree of bias $b$ increases (from 0.5 to 0.9), the performance of the GNN on the biased test set is higher, while the performance on the unbiased test set decreases. This trend illustrates GNNs are more inclined to make predictions with shortcuts compared to the real relationship between the input and label. Finally, based on the above conclusion, these findings further shows GNNs will learn shortcuts during the initial training. We also perform GIN on these datasets in Appendix C.1.

Besides, we conduct additional experiments to show the shortcuts are captured at the early epoch. Specifically, we first conduct experiments on the Spurious-Motif (*Cycle-Tree*). As described in Section 3.1, the *Cycle-Tree* dataset consists of the balance data and the bias data, where bias data contains *Cycle* motifs accompanied by *Tree* bases. As shown in Figure 3(b), we present the training loss curve for the balance/bias example. From the observations, we find that after 2 epochs of training, the training loss of the bias examples is almost zero and the loss of balance data is not converged, indicating that the bias examples are much easier to learn than balance examples in training.

Meanwhile, we show the variation of Accuracy of balance/bias examples during the training phase. From Figure 3(c), we find Accuracy of bias examples achieves high performance early in the training, while the balance examples require more epochs of training to achieve it. This is similar to the observation in Figure 3(b). More experimental results on synthetic datasets are presented in Appendix C.2.

Table 1: The graph classification ACC on testing datasets of the Spurious-Motif and Graph-SST2.

| | | Spurious-Motif | | | | Graph-SST2 |
| | | b=0.5 | b=0.7 | b=0.9 | *Cycle-Tree* | |
|---|---|---|---|---|---|---|
| GIN is the backbone | GIN | $0.3950 \pm 0.0471$ | $0.3872 \pm 0.0531$ | $0.3768 \pm 0.0447$ | $0.3736 \pm 0.0270$ | $0.8269 \pm 0.0259$ |
| | DIR | $0.4444 \pm 0.0621$ | $0.4891 \pm 0.0761$ | $0.4131 \pm 0.0652$ | $0.4039 \pm 0.0425$ | $0.8083 \pm 0.0115$ |
| | DisC | $0.4585 \pm 0.0660$ | $0.4885 \pm 0.1154$ | $0.3859 \pm 0.0400$ | $0.4882 \pm 0.1007$ | $0.8279 \pm 0.0081$ |
| | GERA | $0.4251 \pm 0.0458$ | $0.5331 \pm 0.1509$ | $0.4568 \pm 0.0779$ | $0.3702 \pm 0.0223$ | $0.8301 \pm 0.0088$ |
| | CAL | $0.4734 \pm 0.0681$ | $0.5541 \pm 0.0323$ | $0.4474 \pm 0.0128$ | $0.4362 \pm 0.0642$ | $0.8181 \pm 0.0094$ |
| | GSAT | $0.4517 \pm 0.0422$ | $0.5567 \pm 0.0458$ | $\mathbf{0.4732 \pm 0.0367}$ | $0.3769 \pm 0.0108$ | $0.8272 \pm 0.0064$ |
| | DARE | $0.4843 \pm 0.1080$ | $0.4002 \pm 0.0404$ | $0.4331 \pm 0.0631$ | $0.4527 \pm 0.0562$ | $0.8320 \pm 0.0086$ |
| | SGR | $\mathbf{0.4941 \pm 0.0968}$ | $\mathbf{0.5686 \pm 0.1211}$ | $0.4658 \pm 0.0672$ | $\mathbf{0.5801 \pm 0.1264}$ | $\mathbf{0.8386 \pm 0.0077}$ |
| GCN is the backbone | GCN | $0.4091 \pm 0.0398$ | $0.3772 \pm 0.0763$ | $0.3566 \pm 0.0323$ | $0.3712 \pm 0.0012$ | $0.8208 \pm 0.0165$ |
| | DIR | $0.4281 \pm 0.0520$ | $0.4471 \pm 0.0312$ | $0.4588 \pm 0.0840$ | $0.4325 \pm 0.0583$ | $0.8012 \pm 0.0016$ |
| | DisC | $0.4698 \pm 0.0408$ | $0.4312 \pm 0.0358$ | $0.4713 \pm 0.1390$ | $0.5058 \pm 0.0476$ | $0.8318 \pm 0.0105$ |
| | GERA | $0.4687 \pm 0.0855$ | $0.5467 \pm 0.0742$ | $0.4651 \pm 0.0881$ | $0.5173 \pm 0.0972$ | $0.8269 \pm 0.0077$ |
| | CAL | $0.4245 \pm 0.0152$ | $0.4355 \pm 0.0278$ | $0.3654 \pm 0.0064$ | $0.4593 \pm 0.0489$ | $0.8127 \pm 0.0077$ |
| | GSAT | $0.3630 \pm 0.0444$ | $0.3601 \pm 0.0419$ | $0.3929 \pm 0.0289$ | $0.3474 \pm 0.0031$ | $0.8342 \pm 0.0017$ |
| | DARE | $0.4609 \pm 0.0648$ | $0.5035 \pm 0.0247$ | $0.4494 \pm 0.0526$ | $0.4576 \pm 0.0737$ | $0.8266 \pm 0.0046$ |
| | SGR | $\mathbf{0.4715 \pm 0.0515}$ | $\mathbf{0.5582 \pm 0.0518}$ | $\mathbf{0.4762 \pm 0.1135}$ | $\mathbf{0.5305 \pm 0.1037}$ | $\mathbf{0.8378 \pm 0.0059}$ |

Table 2: The graph classification ROC-AUC on testing datasets of OGBG.

| | | MolHIV | MolToxCast | MolBACE | MolBBBP | MolSIDER |
|---|---|---|---|---|---|---|
| GIN is the backbone | GIN | $0.7447 \pm 0.0293$ | $0.6521 \pm 0.0172$ | $0.8047 \pm 0.0172$ | $0.6584 \pm 0.0224$ | $0.5977 \pm 0.0176$ |
| | DIR | $0.6303 \pm 0.0607$ | $0.5451 \pm 0.0092$ | $0.7391 \pm 0.0282$ | $0.6460 \pm 0.0139$ | $0.4989 \pm 0.0115$ |
| | DisC | $0.7731 \pm 0.0101$ | $0.6662 \pm 0.0089$ | $0.8293 \pm 0.0171$ | $0.6963 \pm 0.0206$ | $0.5846 \pm 0.0169$ |
| | GERA | $0.7714 \pm 0.0153$ | $0.6694 \pm 0.0043$ | $0.8187 \pm 0.0195$ | $0.6953 \pm 0.0229$ | $0.5864 \pm 0.0052$ |
| | CAL | $0.7339 \pm 0.0077$ | $0.6476 \pm 0.0066$ | $0.7848 \pm 0.0107$ | $0.6582 \pm 0.0397$ | $0.5965 \pm 0.0116$ |
| | GSAT | $0.7524 \pm 0.0166$ | $0.6174 \pm 0.0069$ | $0.7021 \pm 0.0354$ | $0.6722 \pm 0.0197$ | $0.6041 \pm 0.0096$ |
| | DARE | $0.7836 \pm 0.0015$ | $0.6677 \pm 0.0058$ | $0.8239 \pm 0.0192$ | $0.6820 \pm 0.0246$ | $0.5921 \pm 0.0260$ |
| | SGR | $\mathbf{0.7945 \pm 0.0071}$ | $\mathbf{0.6723 \pm 0.0061}$ | $\mathbf{0.8305 \pm 0.0098}$ | $\mathbf{0.7021 \pm 0.0190}$ | $\mathbf{0.6092 \pm 0.0288}$ |
| GCN is the backbone | GCN | $0.7128 \pm 0.0188$ | $0.6497 \pm 0.0114$ | $0.8135 \pm 0.0256$ | $0.6665 \pm 0.0242$ | $0.6108 \pm 0.0075$ |
| | DIR | $0.4258 \pm 0.1084$ | $0.5077 \pm 0.0094$ | $0.7002 \pm 0.0634$ | $0.5069 \pm 0.1099$ | $0.5224 \pm 0.0243$ |
| | DisC | $0.7791 \pm 0.0137$ | $0.6626 \pm 0.0055$ | $0.8104 \pm 0.0202$ | $0.7061 \pm 0.0105$ | $0.6110 \pm 0.0091$ |
| | GERA | $0.7816 \pm 0.0079$ | $0.6622 \pm 0.0045$ | $0.8044 \pm 0.0063$ | $0.6970 \pm 0.0089$ | $0.6133 \pm 0.0239$ |
| | CAL | $0.7501 \pm 0.0094$ | $0.6006 \pm 0.0031$ | $0.7802 \pm 0.0207$ | $0.6635 \pm 0.0257$ | $0.5559 \pm 0.0151$ |
| | GSAT | $0.7598 \pm 0.0085$ | $0.6124 \pm 0.0082$ | $0.7141 \pm 0.0233$ | $0.6437 \pm 0.0082$ | $0.6179 \pm 0.0041$ |
| | DARE | $0.7523 \pm 0.0041$ | $0.6618 \pm 0.0065$ | $0.8066 \pm 0.0178$ | $0.6823 \pm 0.0068$ | $0.6192 \pm 0.0079$ |
| | SGR | $\mathbf{0.7822 \pm 0.0079}$ | $\mathbf{0.6668 \pm 0.0026}$ | $\mathbf{0.8228 \pm 0.0283}$ | $\mathbf{0.7116 \pm 0.0169}$ | $\mathbf{0.6217 \pm 0.0291}$ |

Moreover, we also conduct experiments on the real-world dataset (MolBACE). Since we cannot determine which data belongs to the bias data, we show the effectiveness of the *shortcut guider*. Specifically, we first get the different *shortcut guiders* under the different training epochs from 1 to 10 in the first stage of SGR. Next, the trained *shortcut guiders* are incorporated into the SGR during the second stage, and we show the results in Figure 3(d). From the figure, we can observe that the model's performance initially increases with the number of epochs, indicating that the *shortcut guider* captures shortcut features during the early stages of training. After the epoch exceeds 3, the effectiveness of SGR starts to decrease, which illustrates that the *shortcut guider* gradually changes from capturing shortcut features to rationale features as the training continues. The above observations confirm that shortcut information is more likely to be learned in the early stages of training.

**Overall Performance.** To verify the effectiveness of SGR, we first compare it with several baseline methods on the task prediction, and the relevant results are summarized in Table 1 and Table 2. From the observations, we first find that we have better task prediction performance and generalizability compared to the classical GCN and GIN.

Specifically, on the Spurious-Motif data, all methods are trained on the biased dataset and the results are reported based on the unbiased test set. From the experimental results, it can be found that SGR outperforms these base models by a large margin. Meanwhile, our model consistently performs better than GIN and GCN on both OGDB and Graph-SST2. Among them, SGR gains a 4.98% improvement over GIN and 6.94% improvement over GCN on the MolHIV dataset. Since SGR takes GCN and GIN as the backbone respectively, the experimental results suggest that our proposed method can well help existing GNNs to mitigate the negative impact of bias.

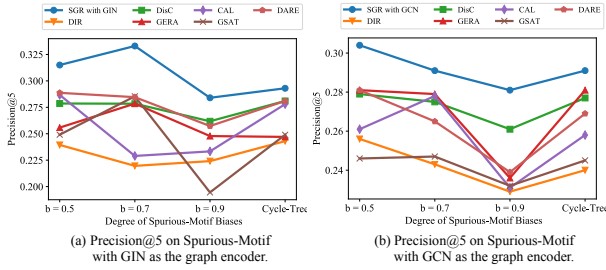

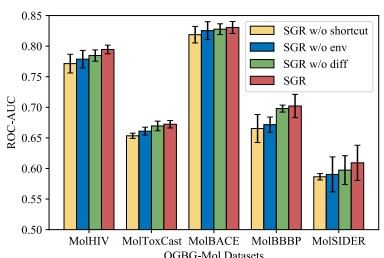

(a) Precision@5 on Spurious-Motif with GIN as the graph encoder.

(b) Precision@5 on Spurious-Motif with GCN as the graph encoder.

Figure 4: The results of identifying the ground-truth rationale subgraphs on Spurious-Motif.

Figure 5: Ablation studies of SGR with GIN over the OGBG dataset.

Then, SGR is also superior to de-biased baselines and performs well on most tasks, indicating the effectiveness of SGR. Among them, we notice that DIR performs poorly on most of the datasets, a possible reason is that it explicitly takes non-rationale subgraphs as environments, which loses some contextual information. In contrast, DisC, GREA and CAL all employ latent non-rationale subgraph representations as environments, and their results are significantly improved compared to DIR. This illustrates the effectiveness of adopting non-rationale representations. SGR still outperforms them, indicating that introducing shortcut in the training phase and allowing the model to learn from shortcut is effective. GSAT does not consider the non-rationale information in data and performs average, which again illustrates the advantage of introducing non-rationale representations. DARE separates the graph as the rationale and non-rationale subgraph by minimizing MI (CLUB_NCE), but DARE does not consider the shortcuts problem in the data, so it is still less effective than SGR.

Finally, to further analyse whether SGR captures invariant rationales, we experiment SGR with the baseline methods on Spurious-Motif that contains the ground-truth rationales, and employ Precision@5 to evaluate the coincidence between the identified rationales and the real ones. Experimental results are shown in Figure 4. From observations, we can find that SGR has an advantage over other methods in finding invariant rationales, regardless of the changing degree of shortcuts in the data.

**Ablation Studies.** To verify the importance of the different components of the model, we construct ablation studies from three aspects: First, we remove the *shortcut guider* (i.e., we ablate $\mathcal{L}_{shortcut}$ in Eq.(10)). We name this variant as SGR w/o shortcut; Second, we remove $\mathcal{L}_{diff}$ (denoted by SGR w/o diff) to verify whether $\mathcal{L}_{diff}$ can make the predictions stable across different environments; Third, we ablate both $\mathcal{L}_e$ and $\mathcal{L}_{diff}$ (SGR w/o env) to demonstrate the effectiveness of non-rationale representations that are considered as environments.

Here, we make ablation studies on the OGBG dataset where SGR is implemented with GIN. As shown in Figure 5, the performance of SGR w/o shortcut decreases rapidly compared to SGR. Without incorporating shortcut information, the performance of SGR w/o shortcut is about similar to some baselines, such as CAL, indicating the effectiveness of learning from shortcut. Besides, we observe that although SGR w/o env only retains $\mathcal{L}_{shortcut}$, it has already exceeded several baselines, again illustrating the significance of our proposed *shortcut guider*. Meanwhile, SGR w/o env is still less effective than the original SGR, which suggests that making non-rationale representations as environments is effective for composing rationales. Finally, we observe that SGR w/o diff performs worse than SGR, dropping by 0.99% on the MolHIV dataset, illustrating that employing $\mathcal{L}_{diff}$ is instructive for identifying invariant rationale.

**Visualizations.** We provide qualitative analyses on the identified rationale subgraphs. First, we present rationales selected by different methods in Figure 6. We train baselines and SGR on the *Cycle-Tree* dataset and visualize a testing example with

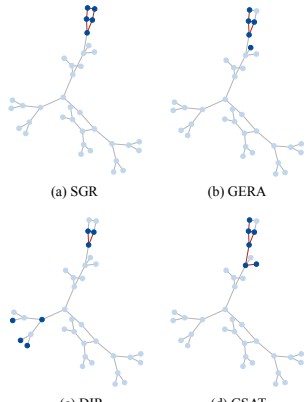

(a) SGR  (b) GERA

(c) DIR  (d) GSAT

Figure 6: Visualization of rationale subgraphs identified by different methods that are trained with the Spurious-Motif dataset *Cycle-Tree*.

motif label as *House*. In Figure 6, the red lines are the edges of rationale subgraph, and the navy blue nodes indicate rationale nodes. Among them, DIR and GSAT identify the edges as rationales. SGR and GERA select the significant nodes as rationales. To make the visualization more intuitive, in SGR and GERA, we assume that if there is an edge between the two identified nodes, we will visualize this

edge as well. From this figure, we can observe that our method can identify more accurate rationales than baselines. More cases can be found in Appendix D.

Besides, we show serval cases of identified rationales for Graph-SST2 on Figure 9 in Appendix D, where Graph-SST2 includes both positive and negative text sentiments. In Figure 9(a)-9(b), we visualize the positive/negative examples in the training set. Among them, we find SGR can accurately highlight some positive tokens ("*the film was better*") in Figure 9(a) and some negative tokens such as "*the opposite of ... magical movie*" in Figure 9(b). Further, we show the effectiveness of our extracted rationale on the OOD test set, where the node degrees in the test set are less than degrees in the training set. For example, it selects "*quite effective*" and "*astonishingly witless*" in Figures 9(c)-9(d) to support the prediction results, respectively. From the above observations, we can conclude that SGR can extract the real rationale subgraph effectively.

## 4 RELATED WORK

**Graph Rationalization.** Graph neural networks (GNNs) on graph classification tasks have achieved remarkable success. However, the prediction results are still unexplainable, rendering most GNNs unreliable. To solve that, Ying et al. (2019); Luo et al. (2020); Yuan et al. (2020); Schlichtkrull et al. (2021) proposed methods to explain the prediction results of GNNs in post-hoc ways, where they explain the predictions of GNNs after they have been trained.

In contrast to these post-hoc methods, recent inherently explainable methods (Veličković et al., 2017; Chen et al., 2022; Li et al., 2022a; Yang et al., 2022) have been investigated for GNNs on graph classification tasks. Among them, graph rationalization methods have been extensively studied. However, recent studies (Chang et al., 2020) have shown that rationalization methods tend to exploit shortcuts in the data to make predictions and compose rationales. To this end, Wu et al. (2022) first proposed to discover invariant rationales by creating multiple environments. They first separated the graph into rationale and non-rationale subgraphs, and explicitly employed the non-rationale subgraphs as environments to identify invariant rationale under environment shifts. Various recent works (Fan et al., 2022; Liu et al., 2022; Sui et al., 2022; Li et al., 2022b) have followed this framework. The difference is that they consider non-rationale subgraph representations as potential environments not the explicit non-rationale subgraph structures. Along another line of research, information bottleneck theory (Tishby et al., 2000; Alemi et al., 2017; Paranjape et al., 2020; Wu et al., 2020; Yu et al., 2021) was introduced into the rationalization. Among them, GSAT (Miao et al., 2022) constrained the information flow from the input graph to the prediction and learned stochasticity-reduced attention to yield rationales. Although most methods are effective in removing shortcuts and discovering rationales, few consider incorporating shortcuts information into the model, enabling the model to learn which information belongs to shortcuts and which is not during the training.

**Shortcut Learning.** Shortcut learning (Geirhos et al., 2020; Du et al., 2022) refers to the phenomenon that deep neural networks (DNNs) highly focus on the spurious correlations in data as shortcuts to predict the results. Although methods with shortcut learning can achieve high performance in identically distributed datasets, it fails to reveal true correlations between the input and the label. When facing the out-of-distribution (OOD) data, the performance will degrade. To solve that, Stacey et al. (2020); Rashid et al. (2021) learned de-biased representations by adversarial training. Arjovsky et al. (2019); Teney et al. (2021); Liu et al. (2022) partition data into different environments and made the prediction under environment shifts. He et al. (2019); Sanh et al. (2021) proposed the product-of-expert method to obtain a de-biased model by a bias-only model.

## 5 CONCLUSION

In this paper, we proposed a shortcut-guided graph rationalization method (SGR) which identified rationale subgraphs by learning from shortcuts. To be specific, SGR involved two stages. In the first stage, a shortcut-only model (*shortcut guider*) was explicitly trained to capture the shortcut information in data with an early stop strategy. During the second stage, SGR separated the input graph into the rationale subgraph representations and the non-rationale ones. Then, the frozen *shortcut guider* was employed to transfer the shortcut information to the above subgraph representations, ensuring the rationale representations could be kept away from the shortcut and the non-rationale ones could encode the same information with shortcuts. Finally, we adopted the non-rationale subgraphs as the environment and then obtained the invariant rationales under environment shifts. Experimental results on both synthetic and real-world datasets demonstrated the effectiveness of SGR.

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

## A  MUTUAL INFORMATION ESTIMATION

In this section, we present the details of InfoNCE and CLUB_NCE used in our SGR.

In probability theory and information theory, the mutual information (MI) of two random variables (e.g., $X$ and $Y$) is a measure of the mutual dependence:

$$I(X;Y) = \mathbb{E}_{p(x,y)} \left[ \log \frac{p(x,y)}{p(x)p(y)} \right]. \tag{11}$$

However, it is hard to calculate the MI values when $X$ and $Y$ are high-dimensional. To solve that, research proposes to estimate the lower and upper bound of MI to achieve MI maximization and minimization. Specifically, for MI maximization tasks, Oord et al. (2018) derive a lower-bound MI estimation (InfoNCE):

$$I_{nce} = \frac{1}{N} \sum_{i=1}^{N} \log \frac{e^{f(x_i,y_i)}}{\frac{1}{N} \sum_{j=1}^{N} e^{f(x_i,y_j)}} = \frac{1}{N} \sum_{i=1}^{N} f(x_i, y_i) - \frac{1}{N} \sum_{i=1}^{N} \left[ \log \frac{1}{N} \sum_{j=1}^{N} e^{f(x_i,y_j)} \right], \tag{12}$$

where $\{(x_i, y_i)\}_{i=1}^{N}$ represents a batch of sample pairs of $(X, Y)$.

For MI minimization tasks, Cheng et al. (2020a;b) propose a Contrastive Log-ratio Upper Bound (CLUB) method:

$$I_{club} = \frac{1}{N} \sum_{i=1}^{N} \log p(y_i|x_i) - \frac{1}{N^2} \sum_{i=1}^{N} \sum_{j=1}^{N} \log p(y_j|x_i), \tag{13}$$

where $p(y|x)$ is a conditional distribution. Further, Yue et al. (2022) develop a new MI minimization method CLUB_NCE which combines InfoNCE and CLUB. CLUB_NCE first adopts the trained $f(x,y)$ by InfoNCE to replace $\log(p(y \mid x))$ in CLUB. Then, it calculates the value of $I_{club}$ based on the trained $f(x,y)$ and minimizes $I_{club}$ to achieve MI minimization. Detailed description of CLUB_NCE can be found in Yue et al. (2022).

## B  EXPERIMENTAL SETUP

### B.1  HOW TO DECIDE THE EPOCH OF THE EARLY STOP STRATEGY?

When selecting the epoch of the early stop strategy, we first define the epoch in the range of [1,5]. Then, for the Spurious-Motif dataset, we choose the epoch as 2 or 3 according to the results in Figure 3(b)-3(c). After that, we perform a grid search to choose the best epoch of the early stop strategy. Based on the experiments, the best epoch for Spurious-Motif is chosen as 2.

Similarly, for Graph-SST2 and OGBG, we choose epochs 3, 4, and 5 based on the results in Figure 3(d). After the grid search, the epoch 3 or 4 is chosen for both Graph-SST2 and OGBG.

### B.2  DATASET STATISTIC

We evaluate our SGR approach on four synthetic datasets from Spurious-Motif [1] (Ying et al., 2019; Wu et al., 2022), and six real-world datasets from Graph-SST2 [2] (Socher et al., 2013; Yuan et al., 2022) and Open Graph Benchmark (OGBG) [3] (Hu et al., 2020). Details of dataset statistics are summarized in Table 3 and Table 4.

---

[1] https://github.com/Wuyxin/DIR-GNN/blob/main/spmotif_gen/spmotif.ipynb
[2] https://github.com/divelab/DIG/tree/main/dig/xgraph/
[3] https://ogb.stanford.edu/docs/graphprop/

Table 3: Statistics of Spurious-Motif and Graph-SST2 Datasets.

| | Spurious-Motif | | | Cycle-Tree | Graph-SST2 |
| | b=0.5 | b=0.7 | b=0.9 | | |
|---|---|---|---|---|---|
| Train/Val/Test | 3,000/3,000/6,000 | 3,000/3,000/6,000 | 3,000/3,000/6,000 | 4,000/4,000/6,000 | 28,327/3,147/12,305 |
| Classes | 3 | 3 | 3 | 3 | 2 |
| Avg. Nodes | 29.6 | 30.8 | 29.4 | 28.9 | 13.7 |
| Avg. Edges | 42.0 | 45.9 | 42.5 | 45.1 | 25.3 |

Table 4: Statistics of OGBG Datasets.

| | MolHIV | MolToxCast | MolBACE | MolBBBP | MolSIDER |
|---|---|---|---|---|---|
| Train/Val/Test | 32,901/4,113/4,113 | 6,860/858/858 | 1,210/151/152 | 1,631/204/204 | 1,141/143/143 |
| Classes | 2 | 617 | 2 | 2 | 27 |
| Avg. Nodes | 25.5 | 18.8 | 34.1 | 24.1 | 33.6 |
| Avg. Edges | 27.5 | 19.3 | 36.9 | 26.0 | 35.4 |

### B.3 BASELINES

In our experiments, we implement all of explainable baselines (**DIR** [4] (Wu et al., 2022), **DisC** [5] (Fan et al., 2022), **GREA** [6] (Liu et al., 2022), **CAL** [7] (Sui et al., 2022), **GSAT** [8] (Miao et al., 2022), and **DARE** [9] (Yue et al., 2022) ) based on their released codes by employing both GCN (Kipf & Welling, 2017) and GIN (Xu et al., 2019) as the graph encoder, respectively.

## C DO GNNs LEARN SHORTCUTS DURING THE INITIAL TRAINING?

### C.1 ACCURACY OF GIN AND GCN ON THE UNBIASED/BIASED TEST SET

In this section, we show the accuracy of GIN and GCN on the unbiased/biased test set in Figure 7. We observe both GIN and GCN achieve promising results where the accuracy values are almost to 100% on biased test set. However, when testing on the unbiased test set, the performance of GIN and GCN degrades significantly. The above observations suggest that the introduction of shortcuts in the training set may be detrimental and is easier to learn. Then, based on the conclusion "*If the malignant bias is easier to learn than the real relationship between the input and label, the neural network tends to memorize it first.*", the observations further indicates that the model learns shortcuts during initial training.

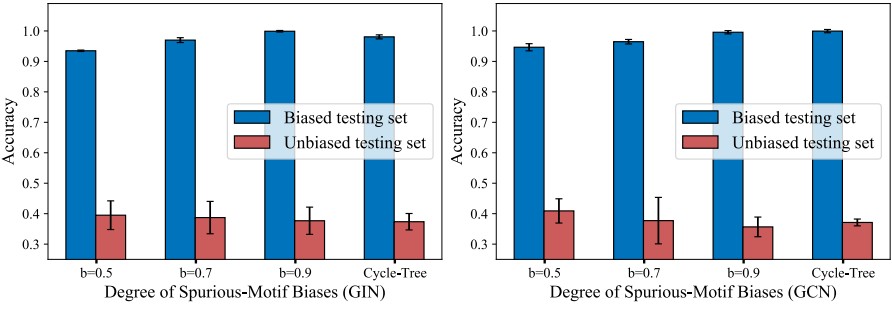

Figure 7: Accuracy of GIN and GCN on the unbiased/biased test set.

---
[4] https://github.com/Wuyxin/DIR-GNN
[5] https://github.com/googlebaba/DisC
[6] https://github.com/liugangcode/GREA
[7] https://github.com/yongduosui/CAL
[8] https://github.com/Graph-COM/GSAT
[9] https://github.com/yuelinan/DARE

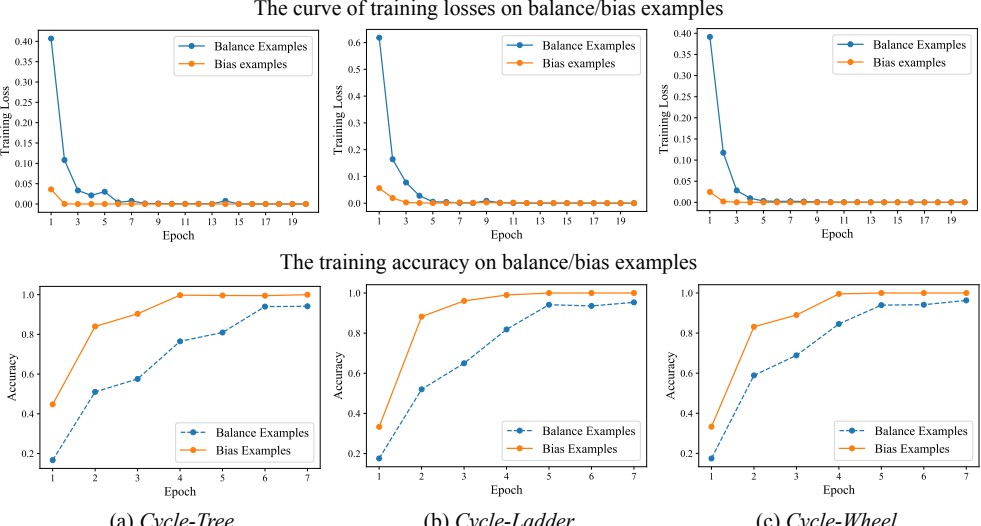

Figure 8: Training losses and accuracy on balance/bias examples with different datasets.

## C.2 CURVE OF TRAINING LOSSES AND TRAINING ACCURACY ON *Cycle-Tree*

In this paper, we assume that shortcut features are easier to learn than the rationale ones. To validate that, we conduct experiments on the synthetic dataset Spurious-Motif (*Cycle-Tree*) which contains both the balance data and the bias data. Among them, the bias data contains *Cycle* motifs accompanied by *Tree* bases. As shown in Figure 8(a), we show the loss curve and accuracy values during training for the balance/bias example. From this figure, we observe the training loss of the bias examples is almost zero when the epoch is 2, while the loss of balance data is not converged. Meanwhile, accuracy of the bias examples achieve high performance early in the training, while the balance examples require more epochs of training to achieve it. The above observations demonstrate that the bias examples are much easier to learn than balance examples in training.

Furthermore, we also conduct experiments on two additional synthetic datasets, where the bias data contains *Cycle* motifs accompanied by *Ladder* bases (denoted by *Cycle-Ladder*), and the bias data contains *Cycle* motifs accompanied by *Wheel* bases (refer to *Cycle-Wheel*). We can observe similar results to Figure 8(a) from Figure 8(b) and 8(c), indicating the bias features are easier to learn.

## D VISUALIZATIONS

We provide some examples of visualizations on the identified rationale subgraphs, including Graph-SST2 (Figure 9) and Spurious-Motif (Figure 10-12):

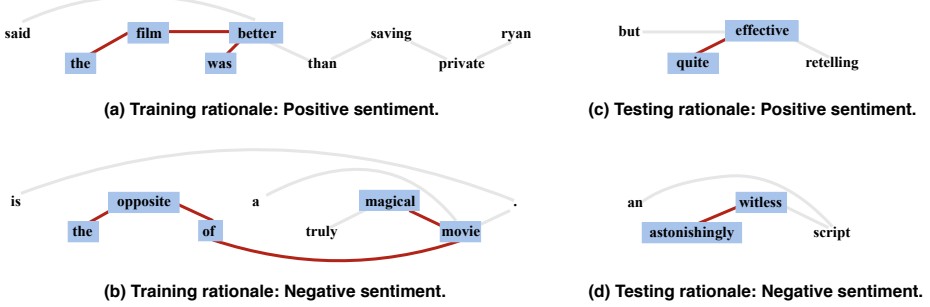

Figure 9: Visualization of SGR rationale subgraphs, where the rationale tokens are highlighted by navy blue colors and the red lines indicate the edges between two identified rationale tokens. Among them, each graph represents a sentiment comment with positive/negative label (e.g., the positive comment "*said the film was better than saving private ryan*" in (a)).

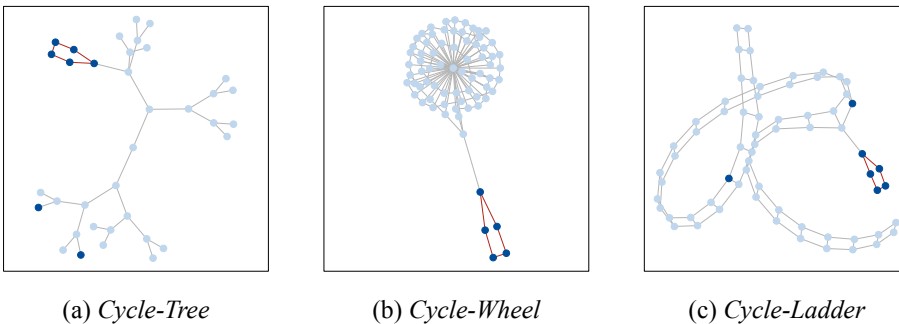

| (a) *Cycle-Tree* | (b) *Cycle-Wheel* | (c) *Cycle-Ladder* |

Figure 10: Visualization of SGR rationale subgraphs, where the selected rationale nodes are highlighted by navy blue colors and the red lines indicate the edges between two identified rationale nodes. Among them, each graph consists of the motif type (*Cycle*) and bases (*Tree*, *Wheel* and *Ladder*).

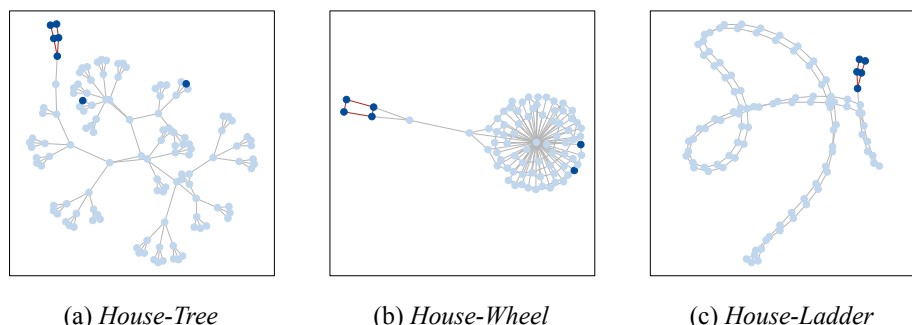

| (a) *House-Tree* | (b) *House-Wheel* | (c) *House-Ladder* |

Figure 11: Visualization of SGR rationale subgraphs, where the selected rationale nodes are highlighted by navy blue colors and the red lines indicate the edges between two identified rationale nodes. Among them, each graph consists of the motif type (*House*) and bases (*Tree*, *Wheel* and *Ladder*).

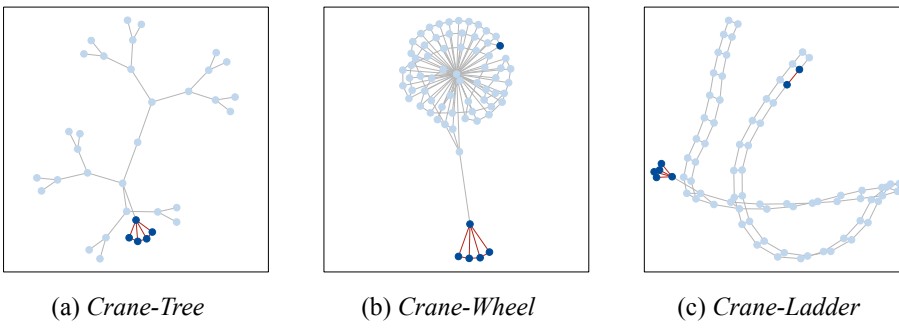

| (a) *Crane-Tree* | (b) *Crane-Wheel* | (c) *Crane-Ladder* |

Figure 12: Visualization of SGR rationale subgraphs, where the selected rationale nodes are highlighted by navy blue colors and the red lines indicate the edges between two identified rationale nodes. Among them, each graph consists of the motif type (*Crane*) and bases (*Tree*, *Wheel* and *Ladder*).

