# OpenReview forum: "Learning from Shortcut: A Shortcut-guided Approach for Graph Rationalization"
_ICLR.cc/2024/Conference — ICLR 2024 Conference Withdrawn Submission_

### Official Review · Reviewer_enA8 · 2023-10-28

**Soundness:** 2 fair
**Presentation:** 2 fair
**Contribution:** 2 fair
**Rating:** 5
**Confidence:** 5

**Summary:**

The study addresses graph rationalization using shortcut guidance. It assumes that the graph neural network (GNN) might become biased towards shortcuts during early training. To counter this, an extra GNN is trained to recognize these shortcuts. This auxiliary GNN helps the main graph rationalization method avoid confusing shortcuts with rationale subgraphs. The method is tested using both simple and real-world molecular graphs.

**Strengths:**

1. The paper focuses on a crucial topic for graph neural networks: interpretability and generalization.

2. The concept of enhancing graph rationalization with shortcut learning is attractive.

3. The experiments seem to validate the promise of the proposed method.

**Weaknesses:**

First, an important assumption mentioned several times in the paper (quoted below) is not well-justified.
> ... previous research (Li et al., 2021; Nam et al., 2020; Fan et al., 2022) suggests that shortcut features are easier to learn than rationale features, indicating that the features learned in the initial training stages are more inclined to shortcuts ...
>
Experiments on the toy datasets (on page 6) show that a higher degree of bias might make the shortcut easier for GCN to learn in a specific example. To what extent can this conclusion be generalized? For instance, does this observation still hold for real graph classification and regression tasks (do they truly suffer from significant bias in model training)? Is there any theoretical proof or more compelling empirical evidence? Do existing graph rationalization methods face the same problems? The authors also cited a few papers here [1,3,4]. [1] focused on the backdoor attack, while the standard training data for graph learning is curated and is intentionally designed to be clean [2]. [3] claimed that "neural networks learn to rely on the spurious correlation only **when it is “easier” to learn** than the desired knowledge" but it remains unclear whether the confounding factors are truly easier for GNN to learn in real-world examples. [4] mostly focused on the graph from images

Second, the motivation behind the model designs is unclear. For Eq. (6), why doesn't the environment affect the task prediction? And why can an environment shift be achieved by simply adding two representation vectors?


Ref.

[1] Anti-Backdoor Learning: Training Clean Models on Poisoned Data. NeurIPS 2021.

[2] MoleculeNet: a benchmark for molecular machine learning. Chemical Science.

[3] Learning from failure: De-biasing classifier from biased classifier. NeurIPS 2020.

[4] Debiasing Graph Neural Networks via Learning Disentangled Causal Substructure. NeurIPS 2022.

**Questions:**

1. In the ablation studies, when we remove the shortcut loss, why does the model performance seem to underperform compared to other graph rationalization methods? Does this imply that other methods can also avoid shortcuts?

2. Is the proposed method applicable to graph regression tasks? Or it is designed for classification?

3. All the rationales in Figure 6 seem similar. What is the advantage of the proposed method?

4. The reported GCN performance of 0.7128 differs from the implementation in the OGBG leaderboard by the OGB official team.

---

### Official Review · Reviewer_LR3y · 2023-10-31

**Soundness:** 2 fair
**Presentation:** 3 good
**Contribution:** 2 fair
**Rating:** 3
**Confidence:** 4

**Summary:**

The paper addresses the problem of spurious explanations in the existing explainers for graph neural networks (GNNs) for graph classification tasks. The proposed method has two steps. First it identifies the spurious explanation based on the assumption that in the initial phases of learning, the model tries to learn the spurious explanations. Using this assumption the method starts with training the GNN model only for a few epochs and the READOUT (e.g.,  global average) layer is used to compute a representation for the spurious data (denoted by h_s). Second, it removes this spurious explanation from the base explanation to improve its quality. The method uses two GNNs to get the node representations and to compute the probability that each of these nodes is in the explanation respectively. By taking the weighted average of the node representations along with the probabilities the method generates the representation for the explanation (denoted by h_r). Then the mutual information between h_s and h_r is minimized to remove the spurious information from the explanations.

**Strengths:**

- The paper is easy to follow.

- The experiments have several baselines and datasets.

- The problem setting is interesting.

**Weaknesses:**

- One major weakness is the assumption that is Spurious connections (shortcuts) are easier to learn than the real input and output relationship. The paper quotes a statement from Arpit et al. (2017); Nam et al. (2020), ‘If the malignant bias is easier to learn than the real relationship between the input and label, the neural network tends to memorize it first’. But this does not mean the malignant bias (shortcut) is always easier to learn. The paper does not provide good justification for extending this principle to the statement that ‘malignant bias is learned in the initial epochs of neural network training’. It is not entirely clear from Section 4.3.

- SGR without the shortcut produces similar results in the ablation study. What’s the justification for it?

- The loss function consists of many hyperparameters. It is difficult to understand what component is contributing to the actual solution. In fact, in the ablation studies, all the variations produce similar results.

- The experimental results in Table 1 shows SGR has slightly improved results in most cases. It is not clear how much the setting plays a role. Are the baseline results also considering their best hyperparameter setting?

**Questions:**

Please see the points in the weakness section.

---

### Official Review · Reviewer_gXJ9 · 2023-11-01

**Soundness:** 3 good
**Presentation:** 3 good
**Contribution:** 2 fair
**Rating:** 5
**Confidence:** 4

**Summary:**

This submission addresses the challenge of explainability in graph neural networks (GNNs). The authors introduce a new method called Shortcut-guided Graph Rationalization (SGR), designed to provide rationale by identifying significant nodes or edges in a graph. The SGR method has two stages: 1) it employs a shortcut guide trained with an early stop strategy to capture shortcut information; 2) it separates the graph into rationale and non-rationale parts and uses mutual information (MI) minimization and maximization to refine the rationale subgraph representations. The method is tested on both synthetic and real-world datasets. The experimental results indicate the effectiveness of the proposed method. The core idea is to allow GNNs to learn from these shortcuts to improve their prediction accuracy and reliability, especially under shifts in the data environment.

**Strengths:**

1. Based on some observations from previous work, the proposed Shortcut-guided Graph Rationalization (SGR) method is new as it identifies significant nodes or edges for rationale.

2. SGR demonstrates robustness across various datasets, including synthetic and real-world scenarios, by filtering out misleading correlations. Its ability to refine rationale subgraph representations through mutual information optimization is beneficial under shifts in the data environment.

3. The paper presents extensive experimental results to validate the effectiveness of the SGR method.

**Weaknesses:**

1. The proposed method assumes that shortcut representations are available, which may not always be true in real-world applications. This assumption could limit the generalizability of the approach to scenarios where such information is not readily identifiable. Furthermore, it is unclear how to construct these shortcuts effectively.

2. The early stop strategy is the key to success for shortcut representations. In Appendix B, the authors provided some simple tricks, but the validation is unclear in a general setting.

3. The shortcuts are challenging to obtain in a general setting since there is a lack of clear definition. The presence and nature of shortcuts can be highly dependent on the specifics of the dataset. In some cases, what constitutes a shortcut might be subtle or complex, making it hard to pinpoint without in-depth analysis.

4. The authors assume the shortcut representation is available. The shortcut feature is easy to learn. These assumptions may be too strong for some applications.

**Questions:**

In addition to the above weaknesses, I have the following questions:

1. I may be wrong, but I was wondering whether the superior performance of the proposed method is due to extra training shortcuts datasets or other factors. If the shortcuts training data contributes more, then it is a little bit unfair to compare the proposed method with non-shortcut methods. The reason I am asking is that it seems the key challenge of the problem is to identify effective shortcuts in a general setting. Once the shortcuts are identified, you can have some advantages if you use them for training the GNN model. Is it the logic?

2. Identifying shortcuts often requires domain knowledge or not.